# Exploring the link between molecular cloud ices and chondritic organic matter in laboratory

G. Danger [1,2,3✉], V. Vinogradoff [1,2✉], M. Matzka[4,5], J-C. Viennet[6], L. Remusat[6], S. Bernard[6], A. Ruf[1], L. Le Sergeant d'Hendecourt[1,2] & P. Schmitt-Kopplin [4,5]

Carbonaceous meteorites are fragments of asteroids rich in organic material. In the forming solar nebula, parent bodies may have accreted organic materials resulting from the evolution of icy grains observed in dense molecular clouds. The major issues of this scenario are the secondary processes having occurred on asteroids, which may have modified the accreted matter. Here, we explore the evolution of organic analogs of protostellar/protoplanetary disk material once accreted and submitted to aqueous alteration at 150 °C. The evolution of molecular compounds during up to 100 days is monitored by high resolution mass spectrometry. We report significant evolution of the molecular families, with the decreases of H/C and N/C ratios. We find that the post-aqueous products share compositional similarities with the soluble organic matter of the Murchison meteorite. These results give a comprehensive scenario of the possible link between carbonaceous meteorites and ices of dense molecular clouds.

[1] Aix-Marseille Université, Laboratoire de Physique des Interactions Ioniques et Moléculaires, UMR 7345, CNRS, Marseille, France. [2] Aix Marseille Université, Laboratoire d'Astrophysique de Marseille, UMR 7326, CNRS, CNES, Marseille, France. [3] Institut Universitaire de France (IUF), Paris, France. [4] Helmholtz Zentrum München, Analytical BioGeoChemistry, Neuherberg, Germany. [5] Technische Universität München, Chair of Analytical Food Chemistry, Freising-Weihenstephan, Germany. [6] Muséum National d'Histoire Naturelle, Sorbonne Université, UMR CNRS 7590, Institut de minéralogie, de physique des matériaux et de cosmochimie, Paris, France. ✉email: gregoire.danger@univ-amu.fr; vassilissa.vinogradoff@univ-amu.fr

Carbonaceous meteorites sampled the organic matter that was available in the solar system at the time of forming planets, including Earth[1–4]. Their organic matter presents a large molecular and structural diversity continuum, divided into two fractions for its characterization: the insoluble organic matter (IOM) and the soluble organic matter (SOM)[5]. Several scenarios were proposed to explain the origin of these organic matters that likely comes from various sources[2,6,7]. The chondritic organics exhibits especially D/H isotopic ratios that relate to a possible formation in cold interstellar environment[8–10] or in the outer part of the protosolar nebula[11,12]. Indeed, during the proto-planetary disk formation, organic materials, inherited from the molecular clouds and sticked on icy-grains are likely submitted to irradiation and thermal alteration due to the rising star[13,14]. Laboratory simulations, conducted to investigate the chemical evolution of ices during this phase showed that a rich chemistry can occur and do form a significant molecular diversity[15–17]. Solid organic residues obtained after ice processing contain thousands of species with masses up to 4000 u[18,19], including a high number of different isomers[20]. Depending on the local environment and the distance to the sun or the central star, some molecules would stay on grains being then agglomerated in cometary and asteroid parent bodies during the accretion phase. Nonetheless, the parent bodies of carbonaceous chondrites have experienced aqueous alteration after the accretion phase[21]. The impact of aqueous alteration on organic matter remains a subject of investigation. Experiments from simple molecules show that the initial signature can be lost to form hundreds of new products and insoluble matters after simulated aqueous alteration[22–27].

Here, we develop experiments to follow the evolution of organic matter, from ice analogs of dense molecular clouds to solar nebula stage and then to its possible modifications in parent bodies of carbonaceous chondrites. We investigate the effects of aqueous alteration on a solid organic residue produced in the laboratory by mimicking the evolution of ice from dense molecular cloud to protosolar nebula. This complex organic matter is called here pre-accretional organic residue. We consider in this study that such residue is an analog of organic matter formed during the protosolar nebula evolution and hence available for the planetesimal accretion. This pre-accretional organic residue is then subjected to aqueous alteration at 150 °C and 5 bars for increasing durations, to simulate the aqueous alteration that occurred in meteorite parent bodies. The molecular profiles of each samples were analyzed by high resolution mass spectrometry to obtain a time dependent evolution of the chemical composition of the organic matter. Results from FT-ICR mass spectrometry analyses show rapid molecular evolution of the molecular families, with a decrease of H/C and N/C ratios. In addition, the new molecular distribution of the post-aqueous analog after 100 days is in part similar to the distribution of chemical families in the Murchison SOM (for CHNO and CHO compounds). Aqueous alteration largely transforms the pre-accretional residue. Our study highlights a possible chemical connection between organic matter in carbonaceous meteorites and first molecules originating from molecular cloud ices.

## Results
### Characteristics of the pre-accretional organic residue.
This material, produced from only three simple molecules ($H_2O$, $CH_3OH$, and $NH_3$—see Methods subsections pre-accretionnal residue formation), contains a large molecular diversity. The high resolution achieved by FT-ICR-MS analysis allows to attribute a stoichiometric formula ($C_xH_yO_zN_w$) to ions presenting an exact mass (called molecular attribution in the rest of the manuscript, more detailed in the method subsections FTICR analysis and

Data treatment). Up to 3964 molecular attributions have been determined, scattered over a large range of masses (from 100 to 650 u with an average at 400 u) and compositions, as analyzed in the negative electrospray ionization mode mass spectrometry (Fig. 1). While the initial ice exhibits H/C = 11, O/C = 3, and N/C = 1, these ratios drastically changed to H/C = 1.67, O/C = 0.39, and N/C = 0.22 during the photo and thermal treatments (Table 1). This is consistent with previous residues made from similar ice analogs[18,19]. The corresponding molecular distribution can be defined in three zones in a Van Krevelen diagram. Zone A (Fig. 1A, H/C vs O/C) corresponds to mostly aliphatic compounds (1.25 < H/C < 2) with a moderate number of oxygen atoms (0.1 < O/C < 0.65). Next to these aliphatic molecules, a non-negligible number of compounds presents a higher degree of unsaturation with H/C ranging from 0.5 to 1.25 and poor in oxygen (0.05 < O/C < 0.25). This subpopulation of compounds may correspond to cyclic and heterocyclic compounds with increased oxygen content (Zone B). Another region (Zone C) comprises compounds enriched in O (0.65 < O/C < 0.85) with aliphatic backbones (1.5 < H/C < 2). The H/C vs N/C diagram (Supplementary Fig. 5) shows a similar distribution as observed with the O/C diagram; however, N/C is limited to <0.6. Additional indication of the diversity in the residue composition lies in the number of C, N, and O per molecule that ranges over a large scale, from five to 35 for C, 0 to 10 for N, and 0 to 13 for O, Fig. 2.

### Molecular nature of the post-aqueous organic product.
The aqueous alteration of the pre-accretional organic residue was performed at 150 °C, to be relevant to the alteration temperatures that occurred on chondrites parent bodies[21] (see Methods subsection aqueous alteration experiments). After only 1 day of aqueous alteration, the chemical signature of the residue is dramatically modified with an evolution of all chemical families (Fig. 1B) and then slightly evolves until 100 days of reaction (Table 1). Newly formed molecules present a higher degree of unsaturation resulting from the reorganization of aliphatic moieties. While the maximum mass distribution has decreased (from 400 u towards 350 u in Fig. 1), the overall DBE has increased. The molecular attribution density after 100 days of alteration is centered at a high DBE[7,8] (Supplementary Fig. 1), corresponding to molecules with more unsaturation. This trend proceeds all along the experiments up to 100 days (Figs. 1, 2). Average elemental ratios have all decreased, especially for N/C that is divided by 2 (from 0.2 to N/C = 0.1) and H/C that decreases from 1.67 to 1.32 after 100 days of aqueous alteration (Table 1). The O/C remains stable within the error bars (going from 0.39 to 0.32). These trends are confirmed with the chemical signatures as shown in the Van Krevelen diagrams (Fig. 1 and Supplementary Fig. 2). Compared to the pre-accretional organic residue, the molecular attribution distribution in the post-aqueous organic product is concentrated in a restricted space with 0.5 < H/C < 2 and 0.25 < O/C < 0.5. This is corroborated by a decrease of nitrogen atoms number per molecule (Fig. 2B) and a change in the number of oxygen atoms per molecule from CHNO to CHO families (Fig. 2C, D). The zones A and B described in the pre-accretional residue merge due to chemical condensation in the post-aqueous organic products the more oxygenated molecules of zone B disappearing more rapidly with time. The highest impact of the aqueous alteration is finally observed on the nitrogen abundance and distribution in molecules (Table 1). Molecules having nitrogen after 100 days of aqueous alteration present a decrease of their nitrogen content (average 2) per molecules (up to 25% molecular attributions with N = 1, compared to 12% before alteration) (Fig. 2B). Furthermore, the number of molecular attributions for non-nitrogen bearing molecule (N = 0) has also increased, by a factor 2 after the same experimental time.

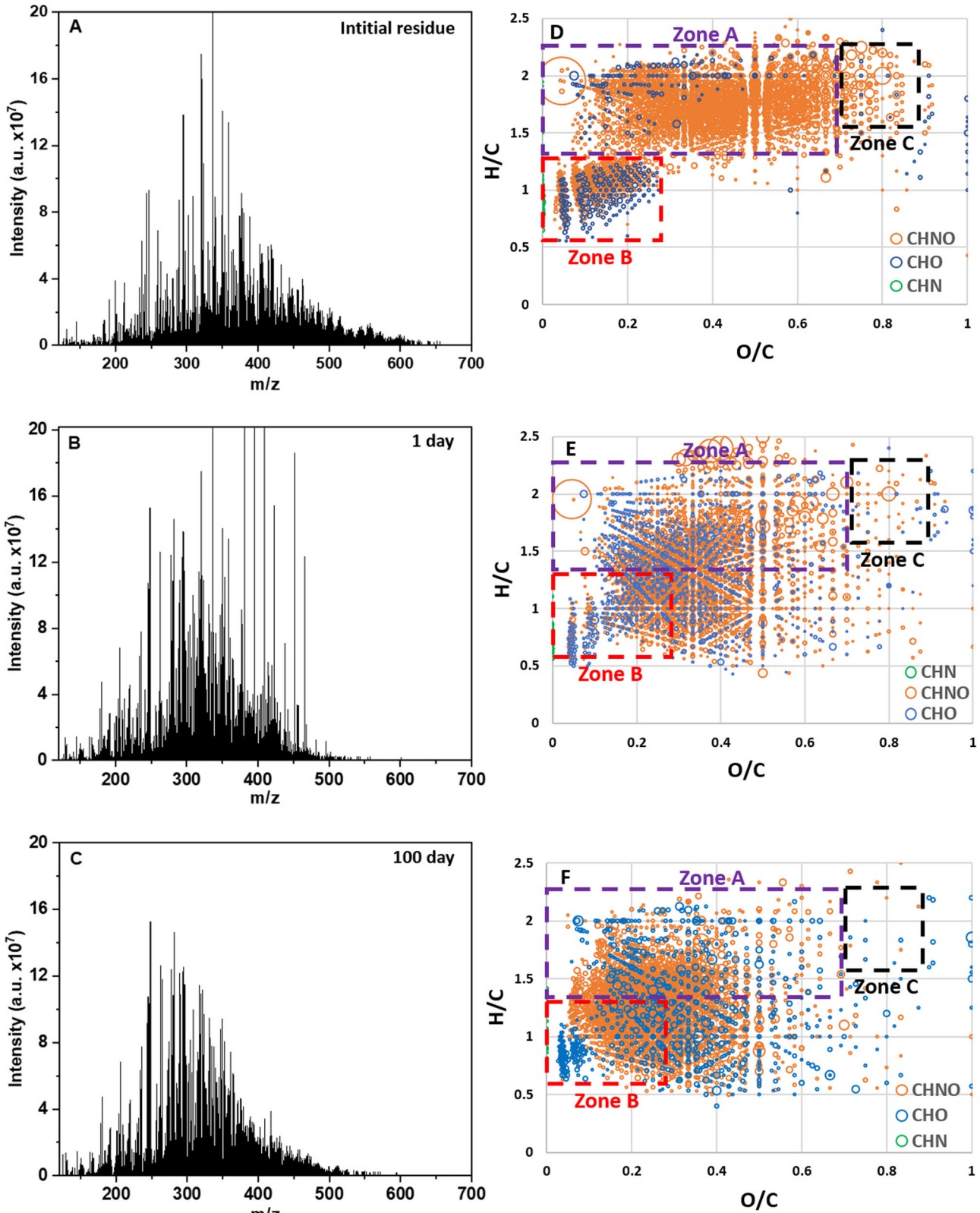

**Fig. 1 Mass spectra and Van Krevelen representations of the samples. A**, **D** The pre-accretional organic residue (3964 molecular attributions), **B**, **E** The post-aqueous organic product after 1 day at 150 °C (3808 molecular attributions), and **C**, **F** The post-aqueous organic product after 100 days (3424 molecular attributions). The size of the circle representing each molecular attribution is proportional to ion intensities. Molecular families of these molecular attributions are colored in green for CHN, orange for CHNO, and blue for CHO. On the Van Krevelen representation are delimitated areas corresponding to aliphatic groups (Zone A), aromatic groups (Zone B), and aliphatic enriched in O groups (Zone C). H/C vs N/C diagrams corresponding to CHNO, CHO, and CHN families separately are displayed in Supplementary Fig. 2–13. The size of the circle representing each molecular attribution is proportional to ion intensities.

**Table 1 Data summary of FT-ICR analyses.**

| Samples | Ice | Pre-accretion | Post-aqueous | | | | Murchison |
|---|---|---|---|---|---|---|---|
| Duration | 0 | 5 | 1 | 10 | 30 | 100 | - |
| Attributions | 3 | 3964 | 3808 | 2301 | 2746 | 3424 | 5661[a] |
| H/C | 11 | 1.67 ± 0.05 | 1.42 ± 0.04 | 1.37 ± 0.04 | 1.40 ± 0.04 | 1.32 ± 0.04 | 1.66[b] |
| O/C | 3 | 0.39 ± 0.05 | 0.34 ± 0.04 | 0.28 ± 0.03 | 0.32 ± 0.04 | 0.32 ± 0.04 | 0.20[b] |
| N/C | 1 | 0.22 ± 0.01 | 0.14 ± 0.01 | 0.13 ± 0.01 | 0.12 ± 0.01 | 0.10 ± 0.01 | 0.03[b] |
| DBE | 0 | 6 | 7 | 7 | 7 | 8 | 8[b] |
| CHNO | - | 3512 | 2739 | 1836 | 2268 | 2591 | 3796 |
| H/C | - | 1.72 ± 0.05 | 1.45 ± 0.04 | 1.38 ± 0.04 | 1.40 ± 0.04 | 1.34 ± 0.04 | 1.45 |
| O/C | - | 0.40 ± 0.05 | 0.35 ± 0.04 | 0.27 ± 0.03 | 0.31 ± 0.04 | 0.32 ± 0.04 | 0.32 |
| N/C | - | 0.16 ± 0.01 | 0.12 ± 0.01 | 0.11 ± 0.01 | 0.11 ± 0.01 | 0.10 ± 0.01 | 0.06 |
| CHO | - | 351 | 1021 | 407 | 472 | 771 | 1684 |
| H/C | - | 1.39 ± 0.04 | 1.32 ± 0.04 | 1.29 ± 0.04 | 1.41 ± 0.04 | 1.26 ± 0.04 | 1.81 |
| O/C | - | 0.25 ± 0.03 | 0.33 ± 0.04 | 0.36 ± 0.04 | 0.42 ± 0.05 | 0.33 ± 0.04 | 0.14 |
| CHN | - | 63 | 22 | 28 | 6 | 15 | 165 |
| H/C | - | 0.94 ± 0.03 | 0.72 ± 0.04 | 1.20 ± 0.07 | 1.36 ± 0.08 | 1.10 ± 0.03 | 1.61 |
| N/C | - | 0.089 ± 0.005 | 0.055 ± 0.003 | 0.15 ± 0.01 | 0.26 ± 0.02 | 0.13 ± 0.01 | 0.11 |

Numbers of molecular attributions, average elemental ratios, average double bond equivalent (DBE), and number of molecular attributions per families for the different samples, initial ice, pre-accretional organic residue, and post-aqueous organic products after 1, 10, 30, and 100 days of aqueous alteration at 150 °C compared to Murchison SOM. Elemental ratios are weighted by ion intensities. Errors for elemental ratio of residues are extrapolated from the data in Fresneau et al. 2017[28] assuming no difference between FT-Orbitrap-MS and FT-ICR-MS analyses since for same residues similar elemental ratio are obtained.
[a]only taking into account molecular attributions containing C, H, N, and/ O atoms.
[b]only considering molecular attributions containing C, H, N, and/or O atoms in Murchison SOM data.

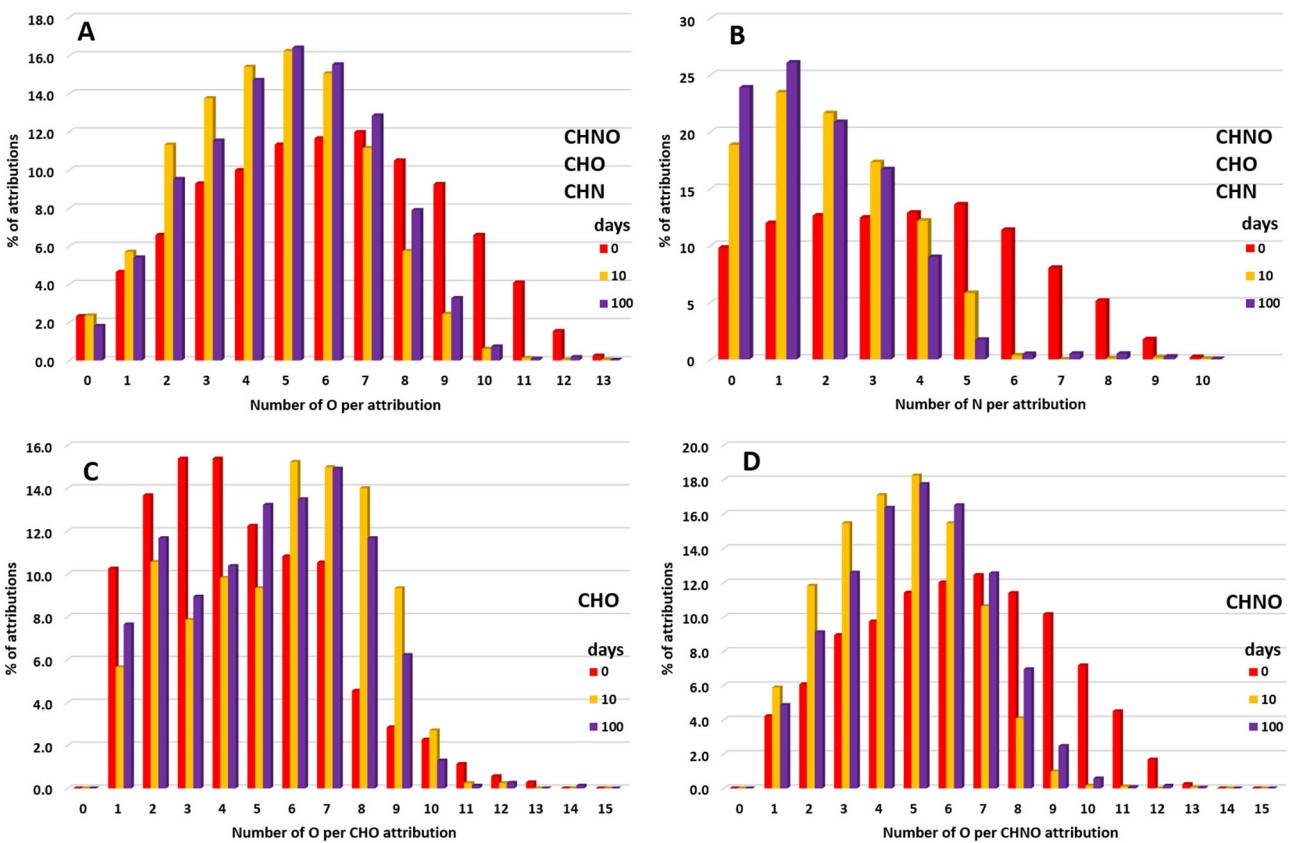

**Fig. 2 Evolution of O and N atoms in Samples.** Histograms of molecular attributions (%) per number of oxygen (**A**) or nitrogen (**B**) for all chemical families (CHO, CHN, and CHNO) for the pre-accretional organic residue and post-aqueous organic products after 10 and 100 days of aqueous reaction at 150 °C. Molecular attributions per number of O for CHO (**C**) or CHNO (**D**) families are also displayed.

**Evolution of chemical families after aqueous alteration.** A more comprehensive view of the chemical modifications can be obtained by focusing on the evolution of the chemical families (CHO, CHNO, and CHN) which composed the samples

(Supplementary Figs. 2–13). The pre-accretional organic residue is mainly composed of CHNO species (89%) followed by CHO (9%) and CHN (2%). However, after 100 days of aqueous alteration, the number of CHNO species have decreased to

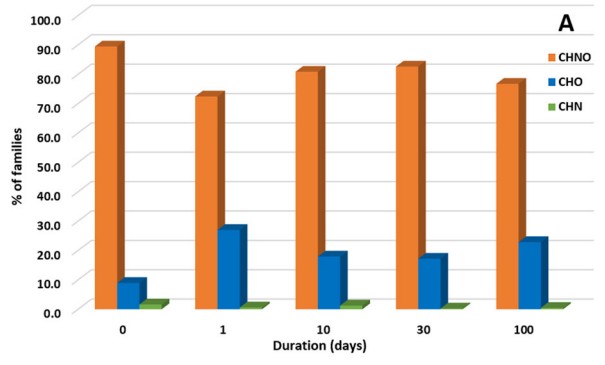

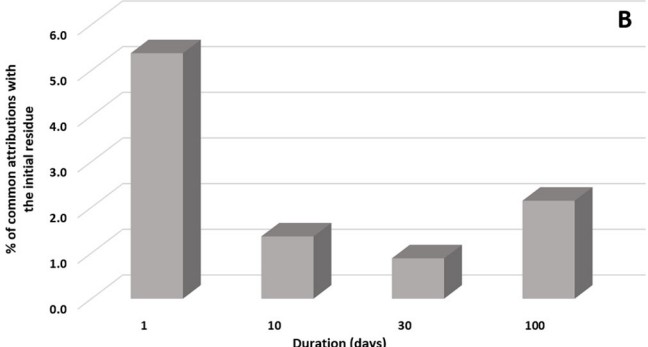

**Fig. 3 Evolution of chemical families and common attributions with the pre-accretional residue.** Percentages of CHNO, CHO, and CHN families (**A**) and of common molecular attributions with the pre-accretional organic analog (**B**) for the different experimental times in the post-aqueous organic products. Common molecular attributions correspond to molecular attributions that are common between the pre-accretional residue and the residue after 1, 10, 30, or 100 days of reaction.

around 77% while the number of CHO species increase up to 23% and CHN species tend to totally disappeared (Fig. 3A).

The evolution of CHNO and CHO species appears to shape the evolution of the pre-accretional organic residue through aqueous alteration, as shown in Fig. 3A. While the CHNO family is distributed over large O/C ratios in the initial organic residue, the range has been lowered after aqueous alteration (Supplementary Figs. 2–7). Evolution of the CHO species shows the opposite trend (Supplementary Figs. 8–10). These trends are confirmed by the evolution of the number of O per molecular attributions for CHO and CHNO families, also related to their respective O/C ratios (Table 1). The aqueous alteration promotes oxidation by increasing the number of O per CHO molecular attributions (Fig. 2C). For the CHNO family, the trend is the opposite, a decrease of the oxygen content is observed with time (Fig. 2D). For these two families (CHNO and CHO), the same evolution is observed for H/C ratios. H/C ratios decrease up to 0.5 to range from 0.5 to 1.5 after 100 days of alteration (Fig. 1), with an average at 1.26 for CHO and 1.34 for CHNO families (Table 1). For the CHNO family, N/C ratios decrease ranging from 0.05 to 0.3 (Supplementary Figs. 5 and 6), with an average at 0.1 (Table 1) after 100 days of aqueous alteration. The CHN species represents only 1.6% of the whole families in the pre-accretional residue and almost totally disappear after 100 days of reaction, since they represent only 0.4 of the whole families. During this evolution, H/C and N/C tend to increase (Table 1), starting from 0.94 and 0.089, respectively to end at 1.10 and 0.13, respectively after 100 days of reaction. The aqueous alteration has thus destroyed CHN species hence affecting significantly their distribution.

## Discussion

**Chemical reactivity of the organic matter generated from ice analogs and subsequently altered under aqueous conditions.** The first step of the organic matter evolution occurs in ice analogs contained in the molecular cloud that collapsed to form the solar nebula[4]. Thanks to laboratory experiments, the basic chemical processes occurring during the photo- and thermal processing of such ice analogs are well-known[16–19,28,29]. The ice photo-process leads to the formation of radicals from the starting molecules that partially recombine at low temperatures[30]. The subsequent warming, simulating the solar nebula evolution, induces thermal processes, leading to at least four types of reactions: acido-basic, nucleophilic, dehydration, and polymerization reactions[16], in addition to desorption of water ice. In laboratory experiments, at 300 K residue of organic matter is observed. In the disk such residue can have stuck on mineral grains. All these reactions likely led to a high molecular diversity, which could have been available in the solid phase. As observed here with the pre-accretional organic residue, more than 10,000 different molecules can be produced, including isomers[20].

During the next step of the protoplanetary disk evolution, this pre-accretional organic material could have been accreted inside parent bodies of carbonaceous meteorites, where second alteration processes have occurred. Under aqueous alteration, our results show that modifications occurred very rapidly in water at 150 °C and 5 bars. After only one day of processing, most of the pre-accretional residue has disappeared (<6 % remaining), highlighting that the pre-accretional organic residue is strongly reprocessed (Fig. 3B). After this period, common molecular attributions with the pre-accretional residue are lower than 2%. It is interesting to note that samples analyzed after 30 and 100 days present up to 10% of molecular attributions in common with each other, while the sample at 10 days present the lowest molecular attributions in common with other samples (Supplementary Fig. 14). After 10 days of experiments a transition seems to occur, where, in the aqueous conditions, new molecules start to form as observed at 30 days of experiments and are still present at 100 days. These observations are consistent with experiments performed on simpler chemical systems, formaldehyde and ammonia or hexamethylenetetramine, which showed rapid transformations of the starting molecules in a few days in aqueous conditions[23,25,31]. As for hexamethylenetetramine[25], we observe here the transformation of nitrogen-bearing compounds through the evolution of N/C values that rapidly decrease after a few days and then stabilize. This decrease of N/C suggests that the most labile N-containing chemical functional groups such as imines, amines, or amides, are the most sensitive to aqueous alteration, with a possible loss of nitrogen, presumably as $NH_3$, $N_2$, and possibly HCN. In contrast, the O/C values of the bulk aqueous products remain stable for all experimental times. Of note, aqueous alteration led to an increase of the O/C ratio when starting from hexamethylenetetramine, while this ratio decreased when starting from formaldehyde/ammonia mixtures. These variations of the O/C in the different analogs, are likely related to the initial chemical composition, which allows hydroxylation or hydrolysis reactions from N-rich molecules compensating dehydration and polymerization reactions from O-rich molecules. CHN molecules have indeed almost completely disappeared in the post-aqueous organic products, along with a decrease of the CHNO molecular attributions. In contrast, a consequent increase of the contribution of the CHO family has been observed. Furthermore, the DBE has also increased with experimental times, clearly implying aromatization of the initial system through dehydration and polymerization/condensation reactions. Overall, the chemical evolution observed in the post-aqueous organic products shows that the organic molecules have promptly

reacted under aqueous alteration via chemical reactions such as aromatization, polymerization, dehydration, and hydroxylation/oxidation reactions. Nonetheless, the diversity after aqueous alteration of the pre-accretional residue remains overall similar (~4000 molecular attributions) while simple molecule systems have produced more than 100 new molecules[23,25,27,31]. It then appears from our experiments that aqueous alteration might not promote molecular diversity when applied to initially complex organic mixtures; it rather exerts a strong control on molecular evolution.

**Possible implications for meteorite organic matter.** The origin of the organic matter in meteorites is a longstanding problem[2,6,7]. The organic materials observed in chondrites could originate from different reservoirs such as organic matter produced by reactions in the nebula gas phase (at elevated temperatures, >600 K)[32] and/or by gas-grain reactions (Fischer–Tropsch processes) at lower temperature (<600 K)[3]. Here, we have investigated the evolution of organic matter formed in ice analogs during solar nebula formation, then supposing its accretion and subsequent processing by aqueous alteration within asteroids.

The average elemental ratios of Murchison SOM methanol extracts, measured with the same analytical technique, are H/C = 1.72, O/C = 0.26, and N/C = 0.03[33], while for the pre-accretional residue they are H/C = 1.67, O/C = 0.39, and N/C = 0.22 (Table 1). The present experiment shows that the aqueous treatment of the pre-accretional residue drastically impacts these ratios that drop to H/C = 1.32, O/C = 0.32, and N/C = 0.10 in the post-aqueous organic products. The aqueous process clearly orients the compositional profiles toward higher aromatic, lower organic matter oxidation states, and lower nitrogen containing compounds (Fig. 4A–C). Aqueous evolution involving aromatization and decrease of the N/C is in agreement with analysis made on organic matter observed within chondrites of different aqueous alteration degree[34,35]. Interestingly, the comparison of the elemental ratios of the different family (CHNO, CHO, and CHN) highlights that the CHO family is the most dissimilar between post-aqueous organic products and Murchison SOM. Murchison CHO family likely contains much longer aliphatic chains and less oxygen groups.

In terms of compositional space on Van Krevelen diagrams, a high coverage for CHO/CHNO families, was found between the post-aqueous organic products and the SOM of Murchison (Fig. 4B, C). The post-aqueous organic products after 100 days share almost half of its CHNO molecular attributions (1200 CHNO, 46%) with CHNO present in Murchison SOM (Supplementary Fig. 16). In comparison, the pre-accretional organic residue is sharing only a third of its CHNO molecules (1000 CHNO, 28%) with Murchison SOM (Supplementary Fig. 15), which is coherent with first observations[19]. For CHO family, 40% of molecular attributions are shared between pre-accretional residue and Murchison SOM, the same trend being observed between post-aqueous products and Murchison SOM. Note that this does not imply strictly identical molecules but molecules with the same number of C, H, O, and N atoms, as different isomers can occur in the experimental samples and Murchison SOM for a same molecular attribution (Supplementary Figs. 15 and 16). Moreover, only 265 CHNO molecular attributions were found common to the pre-accretional organic residue, the post-aqueous organic products, and Murchison SOM. Overall, this demonstrates the strong structural transformations that occurred during the aqueous process (Supplementary Fig. 17). Nonetheless, the 28% common CHNO molecular attributions between the pre-accretion organic residue and Murchison SOM is intriguing. Considering that our pre-accretion organic residue is an analog of

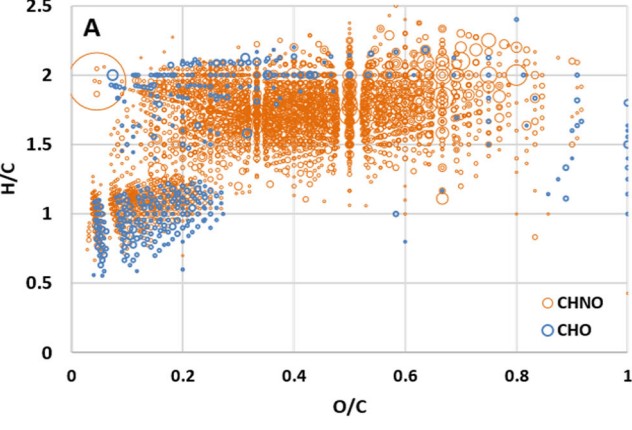

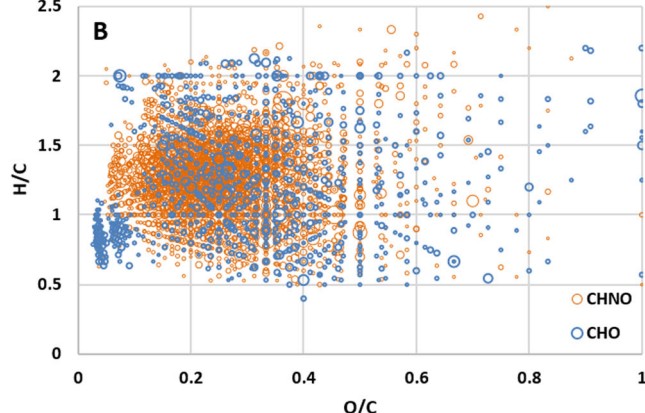

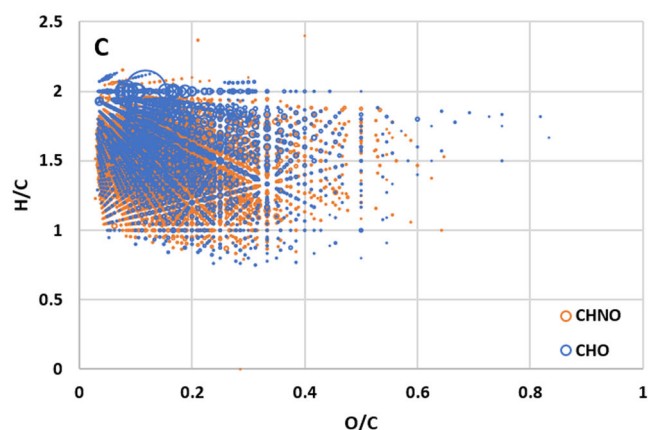

**Fig. 4 Pre-accretional and post-aqueous organic residues compared to the SOM of Murchison.** Figures comparing data from residue (**A**), aqueous alteration after 100 days (**B**), and the CHO/CHNO compositional space of the SOM of the Murchison meteorite (**C**). The size of the circle representing each molecular attribution is proportional to ion intensities.

the organic matter accreted on Murchison parent body, this observation can suggest that the conditions of aqueous alteration in Murchison may have not totally erased the pristine organic signature. Chondrite organic matter shows heterogeneity when investigated in situ at the micron scale[36,37], which can reflect the various local environments that reacted to different degrees during aqueous alteration. Murchison bulk petrological classification was determined at 2.5 by Rubin et al. 2007[38]. However, variations of alteration degree can be observed at the micron-scale. This is likely the consequence of local variations of water to rock ratio that will likely influence the final extend of aqueous

alteration. The extraction of SOM from milligrams of Murchison sample results on average composition that mixes the OM from all these tiny micro-environments. Interestingly, hexamethylenetetramine, a well-known chemical marker observed in pre-accretional organic residue[18,39,40], has been identified in Murchison SOM[41]. This compound has been observed reactive under aqueous alteration, especially at 150 °C[25]. Unfortunately, hexamethylenetetramine is not detectable in the mass range of the FT-ICR-MS analysis, even though it is contained in our pre-accretional organic residue (Danger 2013). Nonetheless, this hexamethylenetetramine detection in Murchison SOM, can indicate that some organic compounds found in chondrites did not suffer too much from the aqueous alteration and were preserved unchanged since their accretion. Therefore, common molecular attributions between our pre-accretional organic residue and Murchison SOM, and between our post-aqueous organic products (at 100 days) and Murchison SOM, impels to continue studying aqueous alteration from such analogs but with different environmental conditions.

Furthermore, some molecules observed in Murchison are not present in organic residues particularly at low O/C and high H/C (Fig. 4C, O/C < 0.1 and H/C > 1.5), and the N/C ratio in residue (0.1) is still higher than observed in Murchison (0.03). This can be due to several factors. Our present laboratory experiments may not totally reflect the temperatures, pressures, and liquid water abundance that may have affected the chemical reactions on the chondrite parent bodies. Other organic precursors, such as gas phase species from the solar nebula[32,42,43] and transformed during the aqueous alteration episode, can also contribute to the organic matter composition observed in carbonaceous chondrites. In addition, strong heating or long duration heating history as well as shock events on the parent body strongly affect the organic structural profiles[44]. Finally, asteroids are essentially composed of mineral matrices and such proximity could have led to complex interactions between minerals and OM during the aqueous alteration phase on asteroids. While the mineral transformation during aqueous alteration is overall well-known[21], its interaction with the organic matter remains discussed[27,36,37]. There was likely a coevolution of the mineral and organic components also involving sulfur chemistry[33] during the aqueous alteration episode in carbonaceous chondrite parent bodies. Also for example, as shown from laboratory experiments, minerals can interact with molecules by changing various chemical pathways due to their trapping inside silica-layers[26,27,45–47] or by influencing the oxidation state of the system[48,49]. In this respect, the impact of mineral and the organo–mineral interactions[50] needs to be also addressed in order to better understand the origin of the SOM in meteorites and its relation to different physico-chemical processes, including environmental ones.

Overall, this laboratory experiment of aqueous alteration on an organic residue produced from interstellar ice analogs clearly demonstrates that aqueous processes can modify the molecular composition of organic matter that could be accreted in parent bodies of chondrites. Considering our scenario, this links the organic matter originating from dense molecular cloud and protoplanetary disks ices with the one found in carbonaceous meteorites. These results tend to confirm that a significant portion of the SOM observed in carbonaceous chondrites can originate from organic ices, inherited from the dense molecular cloud, the progenitor of our solar system.

## Methods
**Pre-acretional residue formation**. A gas mixture of $H_2O:CH_3OH:NH_3$ (2:1:1) was first deposited onto an inert $MgF_2$ window at 77 K to form the ice analog. Irradiation and ice formation by gas deposition were performed simultaneously during

72 h at 77 K using microwave generated $H_2$ plasma with a constant molecular hydrogen flow providing vacuum ultraviolet photons (at Lyman α (121 nm) with a flux of $2 \times 10^{14}$ photons cm$^{-2}$ s$^{-1}$). Ice composition and its irradiation were evaluated with infrared spectroscopy. Twelve residues were made following the same protocol during 3 months. A total of 1.7 mg of organic matter was obtained and kept under argon atmosphere in a stainless steel vessel, to minimize oxidation prior to analysis. We estimate the yield of residue formation of 1.5% w/w compare to the $CH_3OH$ used to their formation. To be recovered, the residue was dissolved in 150 μL pure methanol, completely solubilized, and the methanol was evaporated to obtain the dried residue in a 1 mL vial. The choice of a 2:1:1 ice composition is related to our previous experiments[17,28], showing that such an ice composition is quite a correct benchmark since it represents a large set of residues formed from different ice composition (ranging from 10:1:1 to 3:1:0.2). 77 K simulates the position of an icy grain on the edge of a protoplanetary disk where it can receive a sufficient dose of Lyman α photons.

**Aqueous alteration experiments**. Experiments were conducted in gold capsule (2 cm height, 0.5 cm diameter) inserted in 25 mL Parr reactors. The gold capsules were loaded with 100 μL of the pre-accretional residue dissolved in milli-Q water at a concentration of 1 g/L. Gold capsules were closed under argon atmosphere (>99.999%; Air Liquide, ALPHAGAZ 1) in a glove box (<0.5 ppm $O_2$) and then sealed with an electric arc, out of the glove box, under nitrogen flux. The experimental study was performed in the Parr reactors. The reactors were placed in an oven, at regulated temperature (150 °C) for the different duration. Pressure inside the reactors was not monitored and should correspond to the vapor pressure of water, up to 5 bars at 150 °C. At the end of the experiments, the gold capsules were removed from the reactors and carefully opened under a chemical hood in a clean environment to recover the liquid fraction with a clean syringe in a clean glass vial. Before analyses, samples were kept in fridge, during 2 weeks.

**FT-ICR-MS analyses**. Analyses were performed at the HelmholtzZentrum Muenchen at Munich. Samples were transferred from Paris to Munich at 0 °C to ensure the stability of the samples. Twenty microliters of each sample were diluted in 200 μL methanol (extraction solvent, LC-MS grade; Fluka). The residue of the experimental study was analyzed using a high-field FTICR-MS from Bruker Daltonics with a 12-T magnet from Magnex and parameters described in our previous work[51]. The negative ionization mode was used for the electrospray ionization. A methanol blank was first measured to detect possible solvent contaminations. Each mass spectrum was obtained by accumulating 500 scans in a mass range of 147 to 1000 amu (atomic mass unit).

**Data treatment**. Mass spectra were exported to peak lists at a signal-to-noise ratio 3. Mass resolving power was 400,000 at m/z = 400 with a mass accuracy of <200 ppb, enabling the separate detection of isobars differing by less than the mass of an electron. This allows a direct assignment of molecular compositions with C, H, N, O, and S atoms (and isotopologues in natural abundance) for each individual exact mass (m/z value). These molecular formulas were assigned from exact m/z values by mass difference network analysis for each peak in batch mode by an in-house software tool[52] and validated via the senior-rule approach/cyclomatic number. A $C_xH_yO_zN_wS_v$ molecular attribution then results for each ion, where x, y, z, w, and v are the respective number of atoms. For each data sample, all molecular attributions in common with blanks were discarded from the data set. Only $C_xH_yO_zN_w$ are considered for data treatment, since the initial ice only contains only C, H, N, and O atoms. Since electrospray ionization source is used, only molecules bearing labile hydrogen are easily ionized, implying that only CHN, CHO, and CHNO families are treated in the work. The Double bound equivalent, which corresponds to the number of rings and π bonds in a molecule, was determined using the general formula: $DBE = n(C) - n(H)/2 + n(N)/2 + 1$.

## Data availability
The data that support the findings of this study are available from the corresponding author upon reasonable request.

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

## Acknowledgements
The research was funded with support from the Centre National d'Etudes Spatiales (CNES, R-S18/SU-0003-072 and R-S18/SU-0003- 072, PI: G.D.), and the Centre National de la Recherche Française (CNRS) with the programs Physique et Chimie du Milieu Inter-stellaire (PCMI- PI:G.D.) and Program National de Planétologie (PNP) (PI: V.V., G.D., and L.L.S.d.H.). G.D. is gratefully to the Agence nationale de la recherche for funding via the ANR RAHIIAM-SSOM (ANR-16-CE29-0015). G.D. and A.R. thanks Centre National d'Etudes Spatiales for funding (CNES Postdoctoral Fellowship 2020). L.R. is grateful to the European Research Council for funding via the ERC project HYDROMA (grant agreement No. 819587). M.M. and P.S.-K thank the Deutsche Forschungsgemeinschaft (DFG, German Research Foundation) – Project-ID 364653263 – TRR 235 for their funding.

## Author contributions
G.D., V.V., L.R., and S.B. conceived the present research. G.D., V.V. and J.-C.V. performed the experiments. P.S.-K, M.M., A.R., and G.D. performed FT-ICR-MS analysis and data treatment. G.D. and V.V. wrote the manuscript and all authors contributed to the final version.

## Competing interests
The authors declare no competing interests.
