## [Peer Review File · Nature Communications]

REVIEWER COMMENTS

Reviewer #1 (Remarks to the Author):

In general, this is a well written, clear, and comprehensive paper that gives important new insights in the evolution of complex organic matter under aqueous alteration.

1. What are the noteworthy results?

The paper clearly shows the high diversity that is inferred in protoplanetary nebula ices and how subsequent aqueous alteration leads to the destruction of bands, but importantly the formation of a wealth of new organic bands. These analyses are of unprecedented detail.

2. Will the work be of significance to the field and related fields? How does it compare to the established literature? If the work is not original, please provide relevant references.

This work gives great insight in how the molecular composition of organic residues evolves over time as result of aqueous alteration. The breadth of the molecular composition makes it very relevant for fields including analytical chemistry, astrochemistry and evolution of planetary systems.

3. Does the work support the conclusions and claims, or is additional evidence needed?

The work doe support its conclusions and claims. Clear conclusions are drawn on a level that is supported by the data. Implications are addressed in balanced way for different but related scenarios

4. Are there any flaws in the data analysis, interpretation and conclusions? - Do these prohibit publication or require revision?

I have not observed any flaws. The paper would benefit from a bit more explanation and clarification, but is scientifically very solid.

5. Is the methodology sound? Does the work meet the expected standards in your field?

The methodology is sound and potentially even exceeds expected standards in the fields.

6. Is there enough detail provided in the methods for the work to be reproduced?

The methods are well-detailed. A few things would benefit from clarification, as outlined in the additional comments.

Additional comments on the manuscript

The target audience of this paper is relatively broad, not only analytical chemists. It would be good if you could add 1-2 sentence what you mean with molecular attributions and double bond equivalent (what are the bonds that are equivalent to double bonds). The main point of the paper is the attributions you measure, but for someone not trained in this field, the paper does not make sufficiently clear what these attributions are.

Based on what was the distinction in molecular families made (CHN, CHO, CHNO)? It would be good to have a brief explanation on these families and why no other families (CH fo example)?

L114: non-bearing nitrogen -> do you mean non-nitrogen bearing?

Figure 2: I suspect fig 2A and B show all O bearing attributions and all N bearing attributions in all molecular families you anticipate. The specific mention of CHNO, CHO and CHN gives the impression that other families are also formed that you don't mention. See the related remark above.

The last plot left of Fig S2 is of a different format.

Fig S2 and S3 caption has a grammar error at the second part. <Are displayed ...etc> is not a complete sentence. The sentence occurs again in S4

Can you explain what the sizes of the circles in Figs S1-S4 represent?

I find Fig 3 B a bit vague - what are the common attributions here? I see they decrease and then increase again, but since common is not described it lacks some information for me

L130-140: What happens with the N? You only discuss CHNO and CHO here. CHNO seems to lose N, CHN nearly completely disappears, but you don't describe much of that here, only the decrease in N/C ratios in the CHNO.

L150: It would be good to explain how nearly 4000 attributions relate to over 1000 different molecules.

L152: step of evolution -> molecular evolution? organic evolution?

L154: don't forget to mention the pressure - would the pressure have any influence or would these reactions have taken place at the same rate at 1 bar?

L156, in L155 you say that 6% is left, so your pre accretionary residue is not totally reprocessed.

L160 and remains up to 100 days -> remains what? the new molecules remain up to 100 days? or the reprocessing remains an ongoing process up to 100 days?

When you write up to 100 days, it implies that you've observed a change after 100 days, but if I am correct you stopped the experiments at 100 days, so you don't know what happened after. I would make that clear as well. Do your results suggest that the reprocessing would keep going after 100 days?

L168: what is HMT? I suspect hexamethylenetetramine, so please add the abbreviation at an earlier use of the full name. You don't use HMT anywhere, so you can also remove this one issue and keep using the full name.

L168/9: this is a bit a complex sentence. Do you mean if you started with HMT as starting material the O/C ratio increased over time, but when starting with formaldehyde/ammonia it would have decreased?

L179: have produced new ones -> what does ones mean? attributions? molecules? can you write that out?

L186 : the evolution of an organic matter -> remove an

L222-223: this sentence has grammar issues.

I read: the pre-accretional organic residue is composed of hexamethylenetetramine -> is that correct, or does it contain hexamethylenetetramine?

"is" should go before unfortunately

L225: suffered -> suffer

L226 common formulas -> what kind of formulas

L234: not totally reflect temperatures, pressures, and liquid water abundance, which may —> not totally reflect the temperatures, pressures, and liquid water abundance that may

Residue formation:

Was there still gas left when you started the irradiation? In L255 you say the gas was first deposited onto the MgF2 window.

L258 irradiation was performed simultaneous with what? Or do you mean ice and gas were irradiated simultaneous?

L263 Is this 1700 microgram in total over twelve residues, or per residue?

Aqueous alteration expts:

PARR is a brand name not an acronym: Parr

L269: Cold capsules or gold capsules?

L274: to recovered -> to recover

L274: where did you open the gold capsule?

FT-ICR-MS

L279: experimental study -> the residues of the experimental study were analysed using a high-field...

the experimental study was performed in the Parr reactors.

L280: what are MWords?

Where were the samples analysed? In Schmitt-Kopplin's lab in Germany? If so, how were the samples shipped there? It would be good to mention that, in case you did ship to show you ensured the stability of the samples

Inge Loes ten Kate

Reviewer #2 (Remarks to the Author):

Summary: The water and organic matter entrained in carbonaceous chondrite meteorites, e.g. Murchison, are almost universally agreed upon to be initially derived from molecular constituents contained in icy mantles on interstellar silicate dust grains that ultimately accreted in the protoplanetary disc to form the primitive CC parent body-planetesimal. Direct observation of ice mantles on interstellar dust grains in molecular clouds via the Infrared Space Observatory (ISO) clearly reveals a rich variety of small organic molecules subordinate to H₂O ice that are thought to have been formed primarily through gas grain and ion molecule reactions at very low temperatures.

In the present manuscript the authors describe a series of experiments that seek to connect UV photolysis chemistry of organic containing interstellar ice analogs with what is observed in the soluble inventory of organic molecules extracted from the Murchison carbonaceous chondrite

meteorite. The authors apply extremely high resolution ICR mass spectrometry that exquisitely resolves each molecular constituent with exact molecular formulae. I am always in favor of publication of data rich experimental papers and this paper should be published after a number of points are acknowledged and addressed.

Some issues that require some revision are as follows.

Ice Composition: The experiments employed ices with methanol and ammonia with compositions of 2H₂O:1CH₃OH:1NH₃ or for 100 H₂O molecules, 50 CH₃OH and 50 NH₃, respectively. Direct observation of interstellar ices (via the ISO) reveal that an average composition (over 19 sources) indicate that for 100 H₂O one has (on average) 11.1 CH₃OH and 12.4 NH₃, respectively. The difference between the compositions of the laboratory ices and actual interstellar ices could have significant differences in the molecular products for the following reason. As the authors note UV irradiation at 77 K produces radicals predominantly through C, N, and O-H homolysis. This means at high H₂O content, one will inevitably generate a considerable concentration of hydroxy radicals. The molecules detected in this study arise from the recombination of C and N radical species upon warming up to temperatures near the melting points of the ices. Hydroxy radicals will operate to destroy organic molecules rather than aid in their molecular growth. It is completely expected that using more representative interstellar ice compositions would result in a significant reduction in complex CNO molecules and a reduction in MW across all groups. The authors should acknowledge that the ice compositions chosen are not representative of actual interstellar ices and note that composition will be expected to have a strong control on the molecular products generated.

I am not advocating that the experiments be redone using representative interstellar ice compositions. I would be satisfied with a brief discussion that acknowledges the differences in composition and the role this would play in changing the product yields and molecular distributions.

Reaction Yield: In the methods section it is reported that water:methanol:ammonia ices were deposited on a MgF₂ disk at 77 K. After bringing the disks up to room T (~ 300 K) it was reported that a residue of ~ 1.7 mg was recovered from the surface of the disk. This raises the question as to what was the yield of residue relative to the amount of methanol and ammonia in the initial ice film? I presume the authors know what the thickness of the ices were, so estimates of rxn yield could be ascertained and should be included. This an important piece of information.

Insoluble Residues: Are the residues completely soluble in methanol? In particular after hydrothermal treatment, is there a solvent insoluble residue? This is important because in carbonaceous chondrites, such as Murchison, the vast amount of organic carbon exists as an insoluble complex organic macromolecule. If these experiments are mimicking organic chemical

evolution in a carbonaceous chondrite planetesimal then it would be important to note that an insoluble organic fraction was also generated.

Mass Spectrometry: Much of the mass spectrometry is written for the cognoscenti, for example the term “attribution” is nowhere defined, although I can infer from the text that it corresponds to a unique exact mass formula. All terms should be defined.

Figures 1 and 4 are impossible to read, the corresponding figures in the supplemental material are much clearer, although larger font on the axis labels would be helpful.

Comparison of Murchison relative to experiments: The data in Table 1 provide the easiest means of comparison of the Murchison solubles with the experimental solubles and here we see some significant differences. Whereas the attribution number ranges from ~ 2000 to 4000 in the experiments, it is 13500 in Murchison methanol extract. This appears to be a significant difference. Also, the N/C of the Murchison extract molecules is much lower (N/C=0.03) than the initial residue and lower than the experiments, N/C=0.22 and 0.014-0.10, respectively. Some consideration as to the nature and significance of these differences should be more clearly discussed.

Overall: This is a data rich paper that potentially has relevance to the nature of complex organic matter extractable from carbonaceous chondrite meteorites. The paper should be revised to address the points above before publication.

Responses to reviewer on the Nature Communications manuscript NCOMMS-21-01220-T

From molecular clouds to meteorites: a laboratory simulation of the evolution of soluble organic matter

G. Danger^{1,2,3*}, V. Vinogradoff^{1,2*}, M. Matzka^{4,5}, J-C. Viennet⁶, L. Remusat⁶, S. Bernard⁶, A. Ruf¹, L. Le Sergeant d'Hendecourt^{1,2}, P. Schmitt-Kopplin^{4,5}

We thank the reviewers for their constructive comments that help to clarify the manuscript.

Reviewer #1 (Remarks to the Author):

Additional comments on the manuscript

1. The target audience of this paper is relatively broad, not only analytical chemists. It would be good if you could add 1-2 sentence what you mean with molecular attributions and double bond equivalent (what are the bonds that are equivalent to double bonds). The main point of the paper is the attributions you measure, but for someone not trained in this field, the paper does not make sufficiently clear what these attributions are.

Response: Concerning molecular attributions, in the results section, we added in the main text the following sentence p2 lines 62-64: "The high resolution achieved by FT-ICR-MS analysis allows to attribute a stoichiometric formula ($C_xH_yO_zN_w$) to ions presenting an exact mass (called "molecular attribution" in the rest of the manuscript, more detailed in the method section)." **We also added the following sentence in the method section, p8 lines 299-300:** "This results in a $C_xH_yO_zN_wS_v$ molecular attribution for each ion, where x, y, z, w and v are the respective number of atoms." **Furthermore, we simplified the text by using only molecular attributions, throughout the manuscript.**

Concerning the double bond equivalent, we added the following sentence p12 line 338-339: "The Double bond equivalent, which corresponds to the number of rings and π bonds in a molecule, was determined using the general formula: $DBE = n(C) - n(H)/2 + n(N)/2 + 1$."

2. Based on what was the distinction in molecular families made (CHN, CHO, CHNO)? It would be good to have a brief explanation on these families and why no other families (CH for example)?

Response: We added the following p12 lines 332-337: "This results in a $C_xH_yO_zN_wS_v$ molecular attribution for each ion, where x, y, z, w and v are the respective number of atoms. For each data sample, all attributions in common with blanks were discarded from the data set. Furthermore, due to the initial ice composition, that contains only C, H, N and O atoms, only $C_xH_yO_zN_w$ are considered for data treatment. Since electrospray ionization source is used, only molecules bearing labile hydrogen are easily ionized, implying that only CHN, CHO and CHNO families are treated in the work."

3. L114: non-bearing nitrogen -> do you mean non-nitrogen bearing?

Response: The reviewer is right. Replaced p 4 line 117.

4. Figure 2: I suspect fig 2A and B show all O bearing attributions and all N bearing attributions in all molecular families you anticipate. The specific mention of CHNO, CHO and CHN gives the impression that other families are also formed that you don't mention. See the related remark above.

Response: To limit the misunderstanding, we replace the CHNO, CHO and CHN families by all families in Fig2A and 2B.

5. The last plot left of Fig S2 is of a different format.

Response: Modified.

6. Fig S2 and S3 caption has a grammar error at the second part. is not a complete sentence. The sentence occurs again in S4.

Response: replaced in all captions by: "Van Krevelen diagrams are displayed for H/C vs O/C or vs N/C, as well as DBE vs O/C or N/C, and as O or N vs m/z."

7. Can you explain what the sizes of the circles in Figs S1-S4 represent?

Response: The sizes of circles are proportional to the intensities of the corresponding ions. We added the following in all captions: "The size of the circle representing each molecular attribution is proportional to ion intensities".

8. I find Fig 3 B a bit vague - what are the common attributions here? I see they decrease and then increase again, but since common is not described it lacks some information for me

Response: To clarify this point, we added the following sentence in the caption of Figure 3: "Common attributions correspond to attributions that are present in the pre-accretional residue and in residue after 10, 30 or 100 days of reaction."

9. L130-140: What happens with the N? You only discuss CHNO and CHO here. CHNO seems to lose N, CHN nearly completely disappears, but you don't describe much of that here, only the decrease in N/C ratios in the CHNO.

Response: we added the following comments on the CHN species p8 lines 156-161: "The CHN species represents only 1.6% of the whole families in the pre-accretional residue and almost totally disappear after 100 days of reaction, since they represent only 0.4 of the whole families. During this evolution, H/C and N/C tend to increase (Table 1), starting from 0.94 and 0.089 respectively to end at 1.10 and 0.13 respectively after 100 days of reaction. The aqueous alteration has thus destroyed CHN species hence affecting significantly their distribution."

10. L150: It would be good to explain how nearly 4000 attributions relate to over 1000 different molecules.

Response: From a previous study, we estimated that up to 10 isomers can be present per ion detected with HRMS. By taking a minimum of 3 isomers per ion, we can estimate that at least 12000 different molecules could be present. We added the reference Ruf, A., Poinot, P., Geffroy, C., D'Hendecourt, L. L. S. & Danger, G. Data-driven uplc-orbitrap ms analysis in astrochemistry. *Life* **9, 1–14 (2019) at the end of the sentence.**

11. L152: step of evolution -> molecular evolution? organic evolution?

Response: To clarify, we changed the initial sentence with the following P8 lines 173-174: "During the next step of the protoplanetary disk evolution, this pre-accretional organic material could have been accreted inside."

12. L154: don't forget to mention the pressure - would the pressure have any influence or would these reactions have taken place at the same rate at 1 bar?

Response: the pressure was reported at p8 line 175. Concerning the difference between 1 and 5 bars, this may influence the thermodynamic equilibrium between liquid and gas phases. For instance, we observe a decrease of the nitrogen amount probably due to the release of volatile species including nitrogen. A lower pressure may induce a higher loss of nitrogen due to a higher amount of volatile species in the gas phase. Experiments at 1 bar should be performed to verify this effect.

13. L156, in L155 you say that 6% is left, so your pre accretionary residue is not totally reprocessed.

Response: the reviewer is right. We changed "totally" by "strongly" p 8 line 177.

14. L160 and remains up to 100 days -> remains what? the new molecules remain up to 100 days? or the reprocessing remains an ongoing process up to 100 days?

Response: We modified the sentence P8 line 182:” After 10 days of experiments a transition seems to occur, where, in the aqueous conditions, new molecules start to form as observed at 30 days of experiments and **are still present at 100 days.**”

15. When you write up to 100 days, it implies that you’ve observed a change after 100 days, but if I am correct you stopped the experiments at 100 days, so you don’t know what happened after. I would make that clear as well. Do your results suggest that the reprocessing would keep going after 100 days?

Response: The reviewer is right. We stopped the experiment at 100 days. Therefore, we do not know which evolution would occur after 100 days. We modified the corresponding sentence as described at point 14.

16. L168: what is HMT? I suspect hexamethylenetetramine, so please add the abbreviation at an earlier use of the full name. You don’t use HMT anywhere, so you can also remove this one issue and keep using the full name.

Response: We replaced HMT by hexamethylenetetramine.

17. L168/9: this is a bit a complex sentence. Do you mean if you started with HMT as starting material the O/C ratio increased over time, but when starting with formaldehyde/ammonia it would have decreased?

Response: We rewrote this sentence p8 lines 190-191:” aqueous alteration led to an increase of the O/C ratio when starting from hexamethylenetetramine, while this ratio decreased when starting from formaldehyde/ammonia mixtures”.

18. L179: have produced new ones -> what does ones mean? attributions? molecules? can you write that out?

Response: we replaced “ones” by “molecules” p 8 line 202.

19. L186 : the evolution of an organic matter -> remove an

Response: Done.

20. L222-223: this sentence has grammar issues.

Response: We modified the sentence p9 line 245-247:” Unfortunately, hexamethylenetetramine is not detectable in the mass range of the FT-ICR-MS analysis, even though it is contained in our pre-accretionary organic residue (Danger 2013).”

21. I read: the pre-accretionary organic residue is composed of hexamethylenetetramine -> is that correct, or does it contain hexamethylenetetramine? “is” should go before unfortunately

Response: see point 20.

22. L225: suffered -> suffer

Response: done.

23. L226 common formulas -> what kind of formulas

Response: we changes formulas with molecular attributions p9 line 249.

24. L234: not totally reflect temperatures, pressures, and liquid water abundance, which may —> not totally reflect the temperatures, pressures, and liquid water abundance that may

Response: done.

Residue formation:

25. Was there still gas left when you started the irradiation? In L255 you say the gas was first deposited onto the MgF2 window. L258 irradiation was performed simultaneous with what? Or do you mean ice and gas were irradiated simultaneous?

Response: First the gas is deposited without the irradiation to form the ice and estimate its molecular composition (do we have the 2:1:1 composition?), since the irradiation quickly modify the relative abundance. To clarify this point, we added p 11 line 284-285:" to a dense molecular ice analogue, which composition was evaluated with infrared spectroscopy before irradiation". We also modified p11 line 289:" Irradiation and ice formation by gas deposition were performed simultaneously during 72".

26. L263 Is this 1700 microgram in total over twelve residues, or per residue?

Response: We clarified this point by modifying the sentence p11 line 295:" A total of 1.7 mg of organic matter was obtained".

27. Aqueous alteration expts: PARR is a brand name not an acronym: Parr

Response: done.

28. L269: Cold capsules or gold capsules?

Response: Gold capsules.

29. L274: to recovered -> to recover

Response: done.

30. L274: where did you open the gold capsule?

Response: We added the following p 11 lines 306-307:" carefully opened under a chemical hood in a clean environment to recover the liquid fraction".

FT-ICR-MS

31. L279: experimental study -> the residues of the experimental study were analysed using a high-field...

Response: done p 12 line 312.

32. The experimental study was performed in the Parr reactors.

Response: sentence was modified p 11 line 303:" Gold capsules were closed under argon atmosphere (> 99.999%; Air Liquide, ALPHAGAZ 1) in a glove box (< 0.5 ppm O 2) and then sealed with an electric arc, out of the glove box, under nitrogen flux. The experimental study was performed in Parr reactors."

33. L280: what are MWords?

Response: MWords = Number of data points. We refined the sentence p12 lines 314-315 to "A time domain transient with four MWords (acquisition with 4 million data points) was obtained and Fourier-transformed into a frequency domain spectrum."

34. Where were the samples analysed? In Schmitt-Kopplin's lab in Germany? If so, how were the samples shipped there? It would be go to mention that, in case you did ship to show you ensured the stability of the samples

Response: we added the following sentence p11 lines 308-309:” Analyses were performed at the HelmholtzZentrum Muenchen at Munich. Samples were transferred from Paris to Munich at 0°C to ensure their stability.”

Reviewer #2 (Remarks to the Author):

Summary: The water and organic matter entrained in carbonaceous chondrite meteorites, e.g. Murchison, are almost universally agreed upon to be initially derived from molecular constituents contained in icy mantles on interstellar silicate dust grains that ultimately accreted in the protoplanetary disc to form the primitive CC parent body-planetesimal. Direct observation of ice mantles on interstellar dust grains in molecular clouds via the Infrared Space Observatory (ISO) clearly reveals a rich variety of small organic molecules subordinate to H₂O ice that are thought to have been formed primarily through gas grain and ion molecule reactions at very low temperatures.

In the present manuscript the authors describe a series of experiments that seek to connect UV photolysis chemistry of organic containing interstellar ice analogs with what is observed in the soluble inventory of organic molecules extracted from the Murchison carbonaceous chondrite meteorite. The authors apply extremely high resolution ICR mass spectrometry that exquisitely resolves each molecular constituent with exact molecular formulae. I am always in favor of publication of data rich experimental papers and this paper should be published after a number of points are acknowledged and addressed.

Some issues that require some revision are as follows.

Ice Composition:

1. The experiments employed ices with methanol and ammonia with compositions of 2H₂O:1CH₃OH:1NH₃ or for 100 H₂O molecules, 50 CH₃OH and 50 NH₃, respectively. Direct observation of interstellar ices (via the ISO) reveal that an average composition (over 19 sources) indicate that for 100 H₂O one has (on average) 11.1 CH₃OH and 12.4 NH₃, respectively. The difference between the compositions of the laboratory ices and actual interstellar ices could have significant differences in the molecular products for the following reason. As the authors note UV irradiation at 77 K produces radicals predominantly through C, N, and O-H homolysis. This means at high H₂O content, one will inevitably generate a considerable concentration of hydroxy radicals. The molecules detected in this study arise from the recombination of C and N radical species upon warming up to temperatures near the melting points of the ices. Hydroxy radicals will operate to destroy organic molecules rather than aid in their molecular growth. It is completely expected that using more representative interstellar ice compositions would result in a significant reduction in complex CNO molecules and a reduction in MW across all groups. The authors should acknowledge that the ice compositions chosen are not representative of actual interstellar ices and note that composition will be expected to have a strong control on the molecular products generated. I am not advocating that the experiments be redone using representative interstellar ice compositions. I would be satisfied with a brief discussion that acknowledges the differences in composition and the role this would play in changing the product yields and molecular distributions.

Response: The referee is right because the problem of the ice composition has always been considered in previous works. Indeed, H₂O always dominate the ice composition and some variations do exist in different sources. Although the average over 19 sources is closer to 10/1/1, high mass stars’ sources do reach a ratio 10/3/1. However it seems to be logical to consider ices displaying ratios in accordance to cosmic abundances where depletions in the gas phase are taken into account because ice chemistry in dark clouds will start with these elements and the possibility to easily form hydrides on the grains surfaces. For example, nitrogen is not depleted in the diffuse ISM gas phase whereas O is slightly less depleted than carbon is. Besides, as shown by Bottinelli

et al (2007), most if not all of nitrogen happens to be trapped in NH₃. According to this rule a good ice analogue might then be 2/1.2/0.25 instead of the one presented here 2/1/1. We clearly have an overabundance of N but not C. However, Munoz Caro and Dartois (2009) did not observe strong differences in the infrared features of residues when the amount of H₂O is increased in the initial ice, even if some relative intensity variations are indeed observed. With infrared spectroscopy, they definitely observed an increase of the residue abundance when the amount of water is increased in the initial ice. We confirmed these first observations in 2017 (Fresneau et al 2017) when comparing residues coming from various ice compositions with infrared spectroscopy and high-resolution mass spectrometry. We observed that the molecular diversity measured by high-resolution mass spectrometry is not impacted by the ice composition, and a same mass distribution is observed whatever the ice compositions are within certain limits. Furthermore, we demonstrated that an increase of water in the initial ice leads to an increase of the nitrogen incorporation in the residue, corroborating the XANES analyses performed by Nuevo et al. 2011. For instance, in our experiments, a 10:1:1 gave the same characteristics as a 3:1:5 (H₂O:CH₃OH:NH₃). However, we did not observe an increase of O in residue with water rich ices. A complete discussion about the role of water in residue formation is given in Fresneau et al. 2017, which is out of scope for this paper. Therefore, in these various experiments, authors did not observe a strong destruction of the organic residue when high water contents are present in the initial ice.

Furthermore, in Gautier et al. 2020, we demonstrated that a 3:1:1 (equivalent to a 2:1:1) gave a molecular distribution that encompasses characteristics of residues made from a water rich ice (10:1:1) or NH₃ poor ice (3:1:0.2). Consequently, a residue formed from a 3:1:1 or 2:1:1 ice is quite a satisfactory benchmark to represent a large set of residues formed from different ice molecular ratios. To clarify the choice of our ice composition, we added the following sentence p11 lines 284-286:” The choice of a 2:1:1 ice composition is related to our previous experiments ^{17,28}, showing that such an ice composition is quite a correct benchmark since it represents a large set of residues formed from different ice composition (ranging from 10:1:1 to 3:1:0.2).”

3. Reaction Yield: In the methods section it is reported that water:methanol:ammonia ices were deposited on a MgF₂ disk at 77 K. After bringing the disks up to room T (~ 300 K) it was reported that a residue of ~ 1.7 mg was recovered from the surface of the disk. This raises the question as to what was the yield of residue relative to the amount of methanol and ammonia in the initial ice film? I presume the authors know what the thickness of the ices were, so estimates of rxn yield could be ascertained and should be included. This an important piece of information.

Response: The residue (1.7 mg) was weighed on a balance having a precision of 0.1 µg. Following the reviewer comment, we estimate the yield of the residue formation. For each residue formation, 10 mbar of the ice mixture H₂O:CH₃OH:NH₃ 2:1:1 are used during the experiment. Consequently, 2.5 mbar of CH₃OH are used which correspond to 3x10⁻⁴ mol (reservoir of 3L). Therefore, for 12 residues, the yield is of 1.5% w/w. We added the following sentence p 11 lines 296-297:” We estimate the yield of residue formation of 1.5% w/w compare to the CH₃OH used to their formation.”

Insoluble Residues:

4. Are the residues completely soluble in methanol?

Response: residues are completely soluble in methanol. We added p 8 line 281:” completely solubilized ”

5. In particular after hydrothermal treatment, is there a solvent insoluble residue? This is important because in carbonaceous chondrites, such as Murchison, the vast amount of organic carbon exists as an insoluble complex organic macromolecule. If these experiments are mimicking organic chemical

evolution in a carbonaceous chondrite planetesimal then it would be important to note that an insoluble organic fraction was also generated.

Response: Insoluble organic fraction was not observed after 100 days of reaction.

Mass Spectrometry:

7. Much of the mass spectrometry is written for the cognoscenti, for example the term “attribution” is nowhere defined, although I can infer from the text that it corresponds to a unique exact mass formula. All terms should be defined.

Response: we fixed this point all along the manuscript and added p 2 lines 62-64:” The high resolution achieved by the FT-ICR-MS analysis allows to attribute a stoichiometric formula ($C_xH_yO_zN_w$) to ions presenting an exact mass (called “molecular attribution” in the rest of the manuscript, more detailed in the Methods section). **“ and p 12 lines 332-337:”** This results in a $C_xH_yO_zN_wS_v$ molecular attribution for each ion, where x, y, z, w and v are the respective number of atoms. For each data sample, all attributions in common with blanks were discarded from the data set. Furthermore, due to the initial ice composition that contains only C, H, N and O atoms, only $C_xH_yO_zN_w$ are considered for data treatment. Since electrospray ionization source is used, only molecules bearing labile hydrogen are easily ionized, implying that only CHN, CHO and CHNO families are treated in the work.”

8. Figures 1 and 4 are impossible to read, the corresponding figures in the supplemental material are much clearer, although larger font on the axis labels would be helpful.

Response: We modified the size and resolution of figures 1 and 4.

Comparison of Murchison relative to experiments:

9. The data in Table 1 provide the easiest means of comparison of the Murchison solubles with the experimental solubles and here we see some significant differences. Whereas the attribution number ranges from ~ 2000 to 4000 in the experiments, it is 13500 in Murchison methanol extract. This appears to be a significant difference.

Response: 13500 attributions in Murchison include molecules bearing sulfur (CHOS, CHNOS...). We determined this value by only taking into account CHN, CHO and CHNO families in Murchison. This gives 5661 attributions. We changed this value in Table 1 to avoid confusion.

10. Also, the N/C of the Murchison extract molecules is much lower ($N/C=0.03$) than the initial residue and lower than the experiments, $N/C=0.22$ and $0.014-0.10$, respectively. Some consideration as to the nature and significance of these differences should be more clearly discussed.

Response: As discussed at the end of the manuscript, this difference can be due to the experimental conditions used. We are focusing here on the evolution, even if the value at 100 days is not the same as the one of Murchison. Our experiment clearly demonstrated that the aqueous processing induces a nitrogen loss. We added the following sentence p 10 lines 260-261:” and the N/C ratio in residue (0.1) is still higher than observed in Murchison (0.03).”

REVIEWERS' COMMENTS

Reviewer #1 (Remarks to the Author):

Thank you for addressing and taking into account all the points in my review, I have no further objections to this paper being accepted and published.

Responses to reviewer on the Nature Communications manuscript NCOMMS-21-01220-T

From molecular clouds to meteorites: a laboratory simulation of the evolution of soluble organic matter

G. Danger ^{1,2,3*}, V. Vinogradoff ^{1,2*}, M. Matzka ^{4,5}, J-C. Viennet ⁶, L. Remusat ⁶, S. Bernard ⁶, A. Ruf ¹, L. Le Sergeant d'Hendecourt ^{1,2}, P. Schmitt-Kopplin ^{4,5}

We thank the reviewers for they constructive comments that help to clarify the manuscript.

Reviewer #1 (Remarks to the Author):

Thank you for addressing and taking into account all the points in my review, I have no further objections to this paper being accepted and published.

Response: No comment.

Reviewer #2 (Remarks to the Author):

Response: No comment.